**Data Availability Statement:** The data are all contained within the paper and Supporting Information files.

**Funding:** The authors received no specific funding for this work.

# A do-it-yourself 3D-printed thoracic spine model for anesthesia resident simulation

**Michelle Han** [ID] [1]*, **Alexandra A. Portnova**[2], **Matthew Lester**[3], **Martha Johnson**[4]

**1** Department of Anesthesiology and Pain Medicine, VA Puget Sound Health Care System, University of Washington, Seattle, Washington, United States of America, **2** Department of Mechanical Engineering, Northwestern University, Evanston, Illinois, United States of America, **3** Department of Mechanical Engineering, University of Washington, Seattle, Washington, United States of America, **4** Department of Anesthesiology and Pain Medicine, University of Washington, Seattle, Washington, United States of America

* mhan2@uw.edu

## Abstract

Central line placement, cricothyroidotomy, and lumbar epidural placement are common procedures for which there are simulators to help trainees learn the procedures. However, a model or a simulator for thoracic epidurals is not commonly used by anesthesia training programs to help teach the procedure. This brief technical report aims to share the design and fabrication process of a low-cost and do-it-yourself (DIY) 3D-printed thoracic spine model. Ten expert anesthesiology attendings and fifteen novice anesthesiology residents practiced with the model and were subsequently surveyed to assess their attitudes towards its fidelity and usefulness as a teaching tool. Responses were recorded with a Likert scale and found to be positive for both groups. Design files and an assembly manual were developed and made public through an open-source website.

## Introduction

Thoracic epidural is a powerful tool that can help provide effective analgesia following major thoracic and abdominal surgeries. Its use has been shown to improve surgical outcomes and is included in Enhanced Recovery After Surgery (ERAS) protocols [1]. Trainees learning the procedure often struggle with familiarizing themselves with the proper anatomy, visualizing the geometry of the needle approach, and developing a feel for the relevant tissues. It has been shown that epidurals are the most difficult procedure for beginning anesthesiology residents to learn compared to endotracheal intubation, arterial line placement, brachial plexus blocks, and spinals [2].

Current hands-on learning methods include: a) performing a thoracic epidural on a patient under the supervision of an attending anesthesiologist and b) practicing with either a high- or low-fidelity model. The most commonly used low-fidelity model utilizes a banana to simulate the gradual loss of resistance [3]. High-fidelity models include realistic models of the entire spine [4,5], partial thoracic spinal models [6], and models that combine a physical interface with a virtual-reality display of needle progression [7]. These models range in price from $225 for a partial thoracic spine without soft tissue to over $6,000 for a combined lumbar and

**Competing interests:** The authors have declared that no competing interests exist.

thoracic epidural model. Models that include soft tissue require its constant replacement, which can cost over $1,500.

Because of the high purchase and maintenance costs associated with high-fidelity models, they are not commonly used in anesthesia training programs despite their commercial availability. In addition, some models lack components, such as soft tissue, necessary for a more versatile application. Previous studies evaluating the efficacy of different epidural training methods have not found any significant difference in acquired manual skills between groups learning epidural placement with high-fidelity models and groups practicing on low-fidelity models [8]. However, groups using training models have shown greater improvement in procedural skills compared to groups not using training models [9].

3D printing, with its low cost and flexibility to modify prints, provides an exceptional solution for anatomic modeling for medical education and simulation. By utilizing magnetic resonance imaging (MRI) and computed tomography (CT) data, physicians are able to create patient-specific models that allow for simulation training and preparation for challenging procedures [10,11]. 3D-printed models are currently available for education and simulation in a variety of specialties, including otolaryngology [12], general surgery [13,14] and cardiothoracic surgery [15]. The use and application of 3D-printed models allows trainees to learn and develop skills for a procedure prior to their first attempt on an actual patient.

Currently, there are two 3D-printed models for neuroaxial block simulation described in the literature [16,17], a novel thoracic spine phantom and a lumbar spine phantom. Both aim to further the development of low-cost, procedure specific models, fabricated with a desk-top printer. The lumbar spine phantom compared favorably to a commercial phantom in terms of fidelity. Its printing files are available online, so that it can be replicated by other users. Our paper describes the only open source 3D-printed model available to practice the thoracic epidural procedure. Also, we offer a manual with step by step instructions, so that a user who is familiar with 3D printing can make it. To our knowledge, ours is the only study to gather feedback data from the primary end-users: anesthesiology trainees.

In this project, a simple, low-cost 3D-printed training model was developed with the aim of helping anesthesiology trainees learn thoracic epidural needle placement[18]. User attitudes toward the model were surveyed.

## Methods

The Institutional Review Board of the University of Washington Human Subjects Division exempted this study from approval. Faculty members and residents of the Department of Anesthesiology and Pain Medicine were invited to participate. Informed consent was implied if the faculty member or resident decided to participate. The individual in the photo in this manuscript has given written informed consent (as outlined in PLOS consent form) to publish this image.

### Model design and fabrication

The model consists of 5 thoracic vertebrae and discs, the ligamentum flavum, and surrounding soft tissue. It is designed to be reusable: the ligamentum flavum and the soft tissue can be replaced between uses to ensure a fresh path for the needle.

Thoracic vertebrae from T7 to T11 were identified as the most useful for a realistic training model (thoracic epidural blocks are commonly placed in this region of the spine). Individual models of the vertebrae and their corresponding discs were taken from an open source computer-aided design (CAD) model of the human spine [19]. The CAD model was based on cadaver CT scan data. Thoracic vertebrae were fabricated using a fused deposition 3D printer

(FlashForge Creator Pro) with a commonly used filament, polylactic acid (PLA) (Fig 1A). During the process of fused deposition, the filament is heated to its melting point and extruded onto a build plate in a desired 2D shape. After the completion of each layer, the build plate is moved down in the vertical plane to allow for extrusion of the next layer, which creates a 3D shape. To simulate the human spine, the corresponding vertebral discs were fabricated using the same method but with a flexible material, thermoplastic polyurethane (TPU, hardness of 90A) (Fig 1A). After fabrication, vertebrae and discs were assembled together with epoxy (Fig 1B). The ligamentum flavum was fabricated with a silicone molding material Oomoo (Oomoo30, Smooth-On Inc., Macungie, PA, USA), selected for its rubbery texture and the ability to pass an epidural needle through it. First, a negative mold for the ligamentum flavum was 3D-printed with PLA. Then, Oomoo was poured into the mold and removed after a six-hour cure time (Fig 1C). The ligamentum flavum was secured to a bubble tea straw (a large diameter plastic straw) with duct tape (Fig 1D) and slid into the vertebral canal (Fig 1E).

Ballistic gel was used to create a cylinder of soft tissue around the spine. This material, developed by the US military, closely simulates the density and viscosity of human muscle tissue and is easily made by mixing gelatin powder with water. The spine model was submerged in a plastic container filled with ballistic gel in liquid form that was then solidified in a refrigerator. Lastly, to ensure that the model could stand upright during epidural simulation, it was secured to a wooden stand with a large screw that was placed through the T11 vertebra. The final assembly of the model is shown in Fig 2.

## Survey study design

A survey was developed for anesthesiology attendings and residents at a university teaching hospital. Attendings are board certified anesthesiologists who teach anesthesia. Residents are anesthesia trainees. The primary goal of the study was to assess the attitudes of expert attendings and novice residents towards the model, with a specific focus on its fidelity and its usefulness as a teaching tool. The secondary goal was to assess anesthesia attendings' opinions regarding which aspects of epidural placement are most challenging for residents.

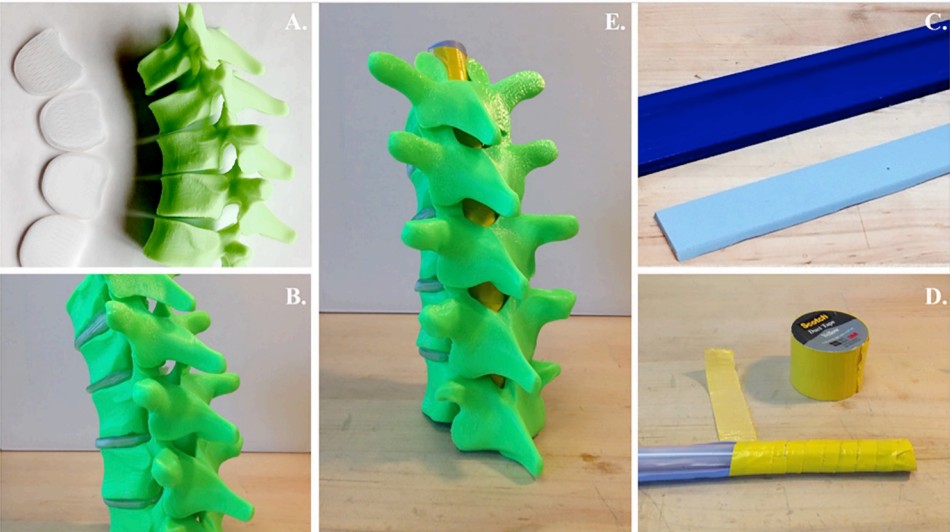

**Fig 1. Steps in model fabrication.** Fabrication of 3D-printed thoracic vertebrae and discs (A), vertebrae and discs epoxied together (B), ligamentum flavum and mold (C), ligamentum flavum secured to bubble tea straw with duct tape (D), ligament and bubble tea straw shown within vertebral canal (E).

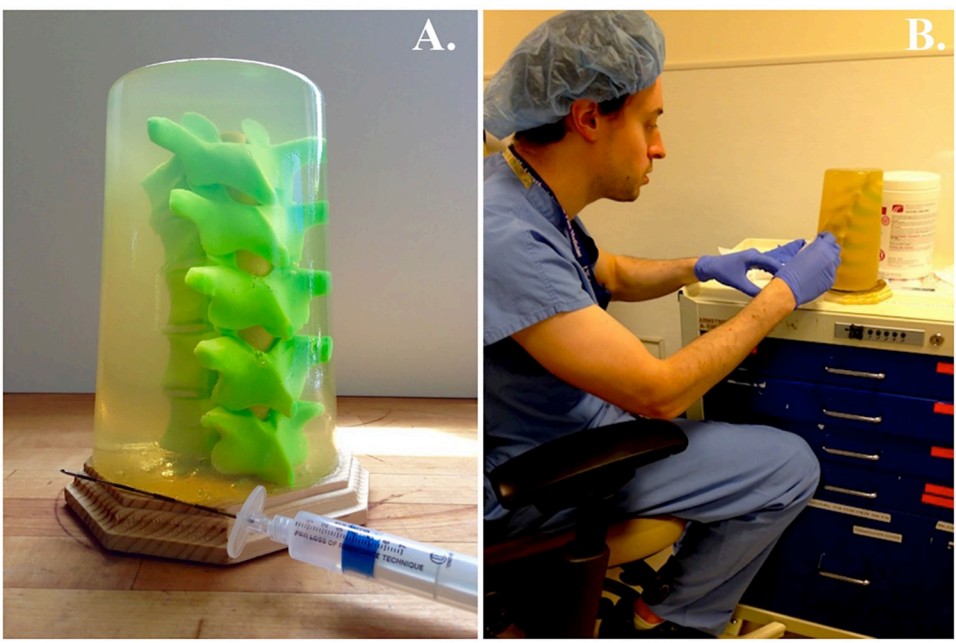

**Fig 2. 3D-printed model.** Thoracic epidural model (A) and trainee practicing thoracic epidural placement (B).

The survey study consisted of two groups: Group A—ten anesthesiology attendings with an average of fifteen years of experience teaching thoracic epidural placement, and Group B—fifteen first year (CA-1) residents who had not yet formally learned thoracic epidural placement. CA-1 residents were chosen in order to have a homogenous group of residents with very little or no experience with thoracic epidurals.

During the study, both groups practiced with the model. Each participant located the epidural space with an epidural needle and a loss-of-resistance syringe using air. In addition, residents were supervised by two of the authors and given basic instructions, similar to what they would receive when placing an epidural needle in a patient for the first time. After practice, both groups filled out a brief anonymous questionnaire to evaluate the model. Attendings were surveyed for: a) their attitudes toward the model, especially its fidelity and its usefulness as a teaching tool, and b) their perception of challenges faced by residents while first learning epidural placement technique. Residents were surveyed for their attitudes toward the model and how it helped them learn the thoracic epidural procedure (Table 1). Both groups were also asked if they would consider making the model themselves. All responses were recorded using a Likert scale.

Lastly, the authors completed ultrasound examination of the model to assess its image quality.

## Results

### Cost and fabrication time

The total cost of the developed model was $40.01. This includes the material costs used in the fabrication, but excludes the potential costs associated with using a 3D printer and purchasing necessary assembly tools. The total fabrication time of the thoracic epidural model was approximately 38.5 hours, with 33 hours 40 minutes being the hands-off time (3D printing and molding) and 4 hours 50 minutes being the hands-on assembly time (Table 2).

**Table 1. Outline of survey questions.**

| | |
|---|---|
| *Attendings' assessment of the fidelity of the model*. | 1. How realistic is the visual representation of the thoracic spine anatomy T7-T11?<br>2. How realistic does the soft tissue feel?<br>3. How realistic does the bone feel?<br>4. How realistic does the ligamentum flavum feel?<br>5. How realistic does the loss of resistance feel?<br>6. How useful do you think the model is as a teaching tool? |
| *Attendings' perspective on the difficulty of various steps for novice residents learning thoracic epidural placement*. | 1. Identifying landmarks on back/identifying different thoracic levels.<br>2. Determining where to place needle on patient's back and determining needle approach.<br>3. How to feel and identify the different layers with needle (soft tissue, bone, ligamentum flavum).<br>4. How to mentally visualize the location of the needle tip with redirection at the skin.<br>5. How to feel the "loss of resistance". |
| *CA-1 Residents' assessment of how the model helps them learn thoracic epidural placement*. | 1. The thoracic epidural model helps me: feel different tissue layers of soft tissue, bone, ligamentum flavum.<br>2. The thoracic epidural model helps me: visualize location of needle tip with redirection at skin surface.<br>3. The thoracic epidural model helps me: improve manual dexterity for epidural needle placement.<br>4. The thoracic epidural model helps me: learn the feeling of loss of resistance.<br>5. The thoracic epidural model helps me: prepare for epidural placement in a patient. |

## Survey results

Attendings' responses to queries regarding the model's fidelity were all positive, ranging from 3.1 (SD = 1.05) for the feel of the soft tissue to 4.8 (SD = 0.41) for the feel of bone. They rated the model's usefulness as a teaching tool as "very good," 4.1 (SD = 0.52) (Fig 3).

Attendings considered "how to mentally visualize the location of the needle tip with redirection at the skin" to be the most challenging aspect of epidural placement for trainees, 3.9 (SD = 1.60). They considered "how to feel the loss of resistance" to be the least challenging aspect, 2.6 (SD = 0.98) (Fig 4).

**Table 2. Cost itemization and construction time for model.**

| Cost Itemization | | Time Breakdown | | | |
|---|---|---|---|---|---|
| | | *Hands-Off Time* | | *Hands-On Time* | |
| PLA filament | $8.18 | | | | |
| TPU filament | $0.59 | Print vertebrae | 11hr 40min | Set up prints | 1hr |
| Gelatin | $8.00 | Print discs | 1hr 30min | Clean prints | 1hr |
| Silicone | $0.51 | Print mold | 2hr | Mix/pour silicone | 20min |
| Plastic mold | $6.66 | Cure silicone | 6hr | Laser cut base | 20min |
| Wooden base | $0.57 | Set Epoxy | 30min | Trim soft tissue mold | 10min |
| Screw | $0.06 | Cure gel | 12hr | Assemble model | 1hr 15min |
| Epoxy | $5.47 | | | Gel model | 45min |
| C Clamp | $9.97 | | | | |
| **Total** | **$40.01** | | **33hr 40min** | | **4hr 50min** |

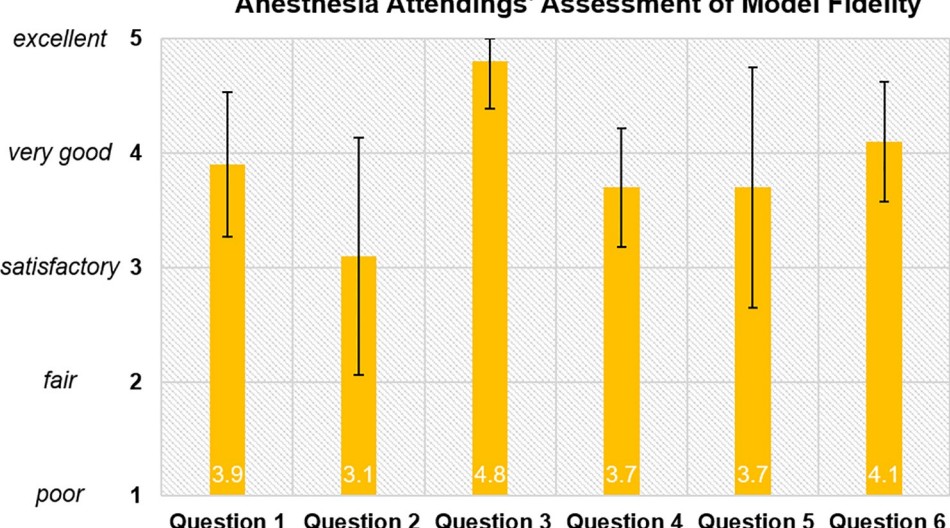

**Fig 3. Anesthesia attendings' assessment of model fidelity.** Question 1. How realistic is the visual representation of the thoracic spine anatomy T7-T11? Question 2. How realistic does the soft tissue feel? Question 3. How realistic does the bone feel? Question 4. How realistic does the ligamentum flavum feel? Question 5. How realistic does the loss of resistance feel? Question 6. How useful do you think the model is as a teaching tool?

Resident responses to queries regarding how the model helps them learn were all positive, ranging from 3.9 (SD = 1.03) for "the thoracic epidural model helps me feel the loss of resistance" to 4.7 (SD = 0.46) for "the thoracic epidural model helps me visualize the needle tip with redirection at skin" (Fig 5). As a group, they were in agreement that the model helped

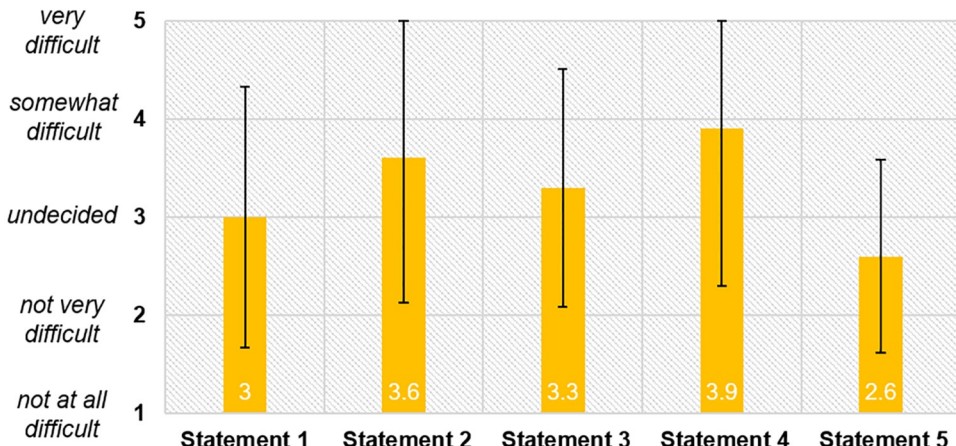

**Fig 4. Anesthesia attendings' perspective on the difficulty of various steps for novice residents learning thoracic epidural placement.** Statement 1. Identifying landmarks on back / identifying different thoracic levels. Statement 2. Determining where to place needle on patient's back and determining needle approach. Statement 3. How to feel and identify the different layers with needle (soft tissue, bone, ligamentum flavum). Statement 4. How to mentally visualize the location of the needle tip with redirection at skin. Statement 5. How to feel the "loss of resistance".

## Anesthesia Residents' Assessment of the Model

strongly agree — 5

agree — 4

neutral — 3

disagree — 2

strongly disagree — 1

| Statement 1 | Statement 2 | Statement 3 | Statement 4 | Statement 5 |
|---|---|---|---|---|
| 4.3 | 4.7 | 4.5 | 3.9 | 4.3 |

**Fig 5. Anesthesia residents' assessment of the model.** Statement 1. The thoracic epidural model helps me feel different tissue layers of soft tissue, bone, ligamentum flavum. Statement 2. The thoracic epidural model helps me visualize location of the needle tip with redirection at skin surface. Statement 3. The thoracic epidural model helps me improve manual dexterity for epidural needle placement. Statement 4. The thoracic epidural model helps me learn the feeling of loss of resistance. Statement 5. The thoracic epidural model helps me prepare for epidural placement in a patient.

them to: feel the different tissue layers (4.3, SD = 0.70), improve manual dexterity (4.5, SD = 0.64), and prepare for epidural placement in a patient (4.3, SD = 0.62).

Anesthesia attendings were more likely than residents to want to construct a model for themselves, 4.2 (SD = 0.75) versus 3.6 (SD = 0.91).

Ultrasound (Sonosite X-porte, HFL50xp/15-6 MSK, Fujifilm USA) of the model revealed the landmarks such as spinous processes, laminae, and ligamentum flavum to be easily identifiable (Fig 6).

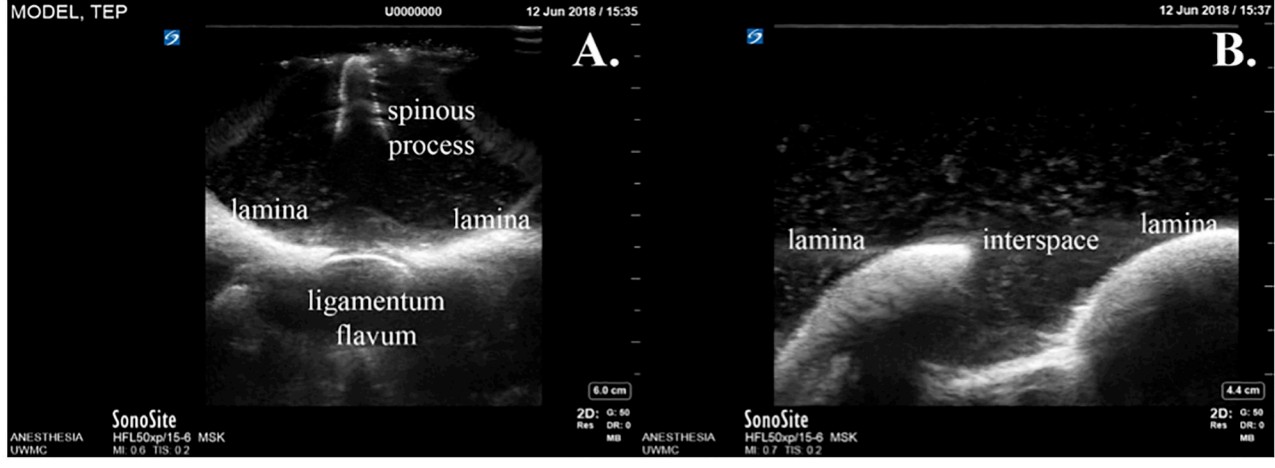

**Fig 6. Ultrasound images obtained from the model.** Transverse view (A) and paramedian view (B).

## Discussion

Procedural competency is typically gained through repeated exposure. A study done by Kopacz showed that most residents obtain proficiency in performing a spinal anesthetic after having performed it approximately 50 times [20]. Another study showed that a novice obtains competency with lumbar epidural placement after 1 to 85 attempts [21]. To learn thoracic epidural placement, trainees can benefit from a simple model with which they can both see and feel the anatomy and practice needle advancement prior to performing the procedure for the first time on an actual patient.

This simple DIY 3D-printed thoracic epidural model is a new educational tool that can be used for anesthesia resident simulation. Similar models have been described previously [16,17] and may help improve the rate at which trainees develop competency, potentially increase the success rate for first attempts at thoracic epidurals, and make the procedure safer for patients.

Survey data of a group of anesthesia experts and novices demonstrates favorable attitudes toward the model. Experts rated the fidelity of its specific features as satisfactory to good, and found its usefulness as a teaching tool to be very good. Novices found it to be a helpful learning tool.

Experts surveyed have found that one of the most challenging steps for novices when learning epidural placement is visualization of the location of the needle tip with redirection at the skin. Novices rated the model favorably in its ability to help them to learn this skill. The transparent gelatin allows for direct visualization of the needle tip in relation to the thoracic spine. The user can thus get a sense for how subtle changes in the angle of the needle approach affect the position of the tip at the depth of lamina and epidural space. This combination of visual and tactile input can be a powerful learning tool [22].

This study has several potential limitations. First, the data set exhibits large standard deviations. Possible explanations include the small sample size and the fact that the responses were recorded with an ordinal scale. A second limitation of the study is the lack of comparison data with traditional teaching methods. Although anesthesia experts' and novices' attitudes toward the model were favorable, the study does not address the question: is it a better means of teaching than a didactic lecture, a small group Problem Based Learning Discussion, or an ultrasound scanning session with a live model? Also, although the cost of the 3D-printed model is significantly less than that of commercially available models, it has not been compared to them in terms of fidelity or educational value. These questions are all areas for future study.

The fabrication time of 3D-printed neuroaxial phantoms is significant. Our print time of approximately 14 hours and hands-on build time of approximately 5 hours compares favorably to other similar models with print times ranging from 25 hours to 3 days, and build times ranging from 6 hours to 6 hours plus an additional day for creating the silicone ligament [16,17]. Although a hands-on build time of 5 hours is significant, this time-cost must be measured against the monetary costs of commercially available models, which are usually in the thousands of dollars. Also, the replaceable ligament and soft tissue components make our model's utility life longer than that of most standard models.

Ballistic gel, although it is inexpensive and provides a satisfactory soft tissue simulation, has some disadvantages. It is an organic material, thus perishable, and degrades after approximately 50 needle passes. The gel can be replaced as necessary and the waste can be composted.

This open source thoracic epidural model can be inexpensively (~$40.00/model) fabricated with a desk-top 3D printer. Our design and material choices were made with the novice builder in mind. In addition, we provide a manual with step by step instructions. Although we do not have global data on who has built the model, to date there have been 1,550 downloads

of our open-source files. We have several copies of the model in our department and it is currently used to teach CA-1 residents the thoracic epidural procedure.

3D printing, with its low cost and ability to produce an almost infinite variety of complex shapes, represents an ideal method for the design and fabrication of medical education models. Current studies of neuroaxial phantoms describe techniques for transforming CT imaging data into files formatted for 3D printing [16,17]. Using these techniques, spine phantoms can be fabricated with normal and pathological variants and thus provide a robust armamentarium of educational tools for neuroaxial anesthesia.

One of the main goals in publishing our model project is for the wider anesthesiology and engineering community to contribute ideas and experiences to further this open access, collaborative effort. Design files and a construction manual can be found at: https://github.com/aport6/thoracic-epidural-model/ and https://www.thingiverse.com/thing:1855444.

## Supporting information

**S1 File. Thoracic epidural model data all data.**
(XLSX)

**S2 File. RAPM Letter.**
(PDF)

## Acknowledgments

The authors would like to thank Ryan Jense, M.D., Vanessa Loland, M.D., and Janet Pavlin, M. D. for their support throughout this project.

## Author Contributions

**Conceptualization:** Martha Johnson.

**Data curation:** Michelle Han.

**Investigation:** Martha Johnson.

**Methodology:** Michelle Han.

**Resources:** Alexandra A. Portnova.

**Software:** Alexandra A. Portnova, Matthew Lester.

**Supervision:** Martha Johnson.

**Writing – original draft:** Michelle Han, Alexandra A. Portnova.

**Writing – review & editing:** Michelle Han, Martha Johnson.

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
