## [Decision Letter · Decision Letter 0]

22 Nov 2019

PONE-D-19-29404

A do-it-yourself 3D-printed thoracic spine model for anesthesia resident simulation

PLOS ONE

Dear Dr. Han,

Thank you for submitting your manuscript to PLOS ONE. After careful consideration, we feel that it has merit but does not fully meet PLOS ONE’s publication criteria as it currently stands. Therefore, we invite you to submit a revised version of the manuscript that addresses the points raised during the review process.

We would appreciate receiving your revised manuscript by Jan 06 2020 11:59PM. To enhance the reproducibility of your results, we recommend that if applicable you deposit your laboratory protocols in protocols.io, where a protocol can be assigned its own identifier (DOI) such that it can be cited independently in the future. For instructions see: http://journals.plos.org/plosone/s/submission-guidelines#loc-laboratory-protocols

We look forward to receiving your revised manuscript.

Kind regards,

Matthew John Meyer, M.D.

Academic Editor

PLOS ONE

Journal Requirements:

2. Thank you for stating in the manuscript Methods:

"University of Washington Human Subjects Division approval was obtained."

b. Once you have amended this/these statement(s) in the Methods section of the manuscript, please add the same text to the “Ethics Statement” field of the submission form (via “Edit Submission”) and clarify whether the study was approved or exempted from approval by your ethical committee.

4. We note that Figure [2] includes an image of a [patient / participant / in the study]. 

Additional Editor Comments (if provided):

Thanks for your submission and creative approach to thoracic epidural teaching. I have a few comments/questions that I would like to be addressed and then I would like you to address the concerns of reviewer #2 before we can accept your manuscript.

1) Citation 16 (Jeganathan et al.) is of a 3d printed model of a spine. Dr. Mashari (one of the reviewers) has published a 3D printed model of the lumbar spine (https://journals.plos.org/plosone/article?id=10.1371/journal.pone.0191664). I would appreciate if you could discuss similarities and the unique/original properties of your project. In order to publish in PLOS ONE, this manuscript "must present a proven advantage over existing alternatives" and I am interested in your opinion as to its "proven advantage over existing alternatives."

2) Both groups were asked if they would make the model themselves--has anyone else made the model for themselves? How challenging would this be for a novice to make the model (if it takes 5hours for an experienced individual, how long for a true rookie)? Have you made more models? Is this currently in use for training UW residents?

3) For figures 3-5, please add the title (which is small and in the upper left corner on my printout) to the actual figure (i.e. Anesthesiologists' assessment of model fidelity).

4) Why did you not include "supraspinous" and "interspinous" ligaments as part of the design?

5) You mention that the ballistic gel is compostable. Would it be possible to make the entire model of compostable materials?

Thanks so much for your generous work towards creating open source teaching models for anesthesiologists.

Sincerely,

Matthew Meyer

Reviewers' comments:

Reviewer's Responses to Questions

**Comments to the Author**

1. Is the manuscript technically sound, and do the data support the conclusions?

Reviewer #1: Yes

Reviewer #2: Partly

2. Has the statistical analysis been performed appropriately and rigorously? 

Reviewer #1: Yes

Reviewer #2: N/A

3. Have the authors made all data underlying the findings in their manuscript fully available?

Reviewer #1: Yes

Reviewer #2: Yes

4. Is the manuscript presented in an intelligible fashion and written in standard English?

Reviewer #1: Yes

Reviewer #2: Yes

5. Review Comments to the Author

Reviewer #1: Well written paper, with complete and succinct description of the training phantom created. Phantom can be replicated with the data and description provided. Evaluation data is provided with adequate detail. Relevant files are available via the Thingiverse website. I would suggest making the files and accompanying documentation also available via version controlled repository (e.g. github or similar) that would allow others to submit potential modifications.

Reviewer #2: This paper is a nice case report on the application of 3D printing for education. The authors then subsequently measured the attitudes of resident learners and attending teachers towards the 3D printed models. The biggest limitation of this study is that it does not actually address the effectiveness of a 3D printed model in education and the attitudes are not relative to a relevant comparator. In other words, there is no comparator group or control group. Generally, any time you give something to people for free or spend any additional educational time, people feel positively about that addition. However, the questions remains, is this better than existing methods of teaching. This study does not answer that question I think. This limitation should be discussed and cited as further work to be performed in the fields.

Although this study has limitations, it is a report of a specific interesting use of 3D printing in medical education.

Specific comments

Pg 2 Ln 43: I am not familiar with the “greengrocer “ terminology, if it is not widely used, would consider removing colloquialisms. And just say easily obtainable fruits and vegetables as models.

pg 3 ln 58 Introduction paragraph: Consider briefly discussing what specific advantages 3D printing offers and why it might be useful for education. Eg cost, flexibility to make different models, complexity of models, etc. Less space on history to keep it concise?

Pg 3 ln 68: Should much of this paragraph be in methods and not introduction since it describes what was done and how?

Pg 4 ln 75: Consider also posting the model to a non-commercial forum such as the NIH 3D print exchange too: https://3dprint.nih.gov/. I believe ThingVerse is owned by Markerbot Industries LLC.

Pg 4 ln 81: Any specific reason this model was chosen? Consider giving credit to the source at University of Iowa, Human Visual Project at https://mri.radiology.uiowa.edu/

Pg 5 ln 115: The study design does not address usefulness or effectiveness as a teaching tool. Rather attitudes of the teachers and learners towards the 3d printed models.

Pg 5 ln 125: Again the survey questions only seem address the attitudes and opinions towards the 3d printed model.

Pg 6 ln 127: I would want to see the actual Likert scale answer choices, and the exact question wording.

Pg 6 ln 128: what was the exact wording of the questions and answers to asking if they would like to make the model themselves.

General Methods comments:

- Was the experience with the model standardized? Ie were specific instructions given? Were the proctors present during usage, were they the same proctors, etcs? How much time was spent with model?

- There is no comparator group or control group. There is no anchoring refence for the survey questions, ie they felt positively about it, but more so than watching a lecture or another traditional method?. How would the model compare to a simple 1 hour didactic or small group PBLD on epidurals or simple scanning on a live model… etc? Generally, any time you give something to people in additional educational time, people feel positively. However, the questions remains, is this Better than existing methods of teaching. This study does not answer that question I think.

Pg 8 ln 160: There appears to be multiple typos in Fig 4 legend.

Pg 9 ln 181: How does the appearance of ultrasound anatomy compare to a real person?

Discussion:

Pg 10 pg 212: this study does not actually who improvement in performance of steps, dexterity or feeling of tissues. Only the self reported attitudes of the learners that may or may not correlate with actually performance.

Pg 10 ln 220: Would discuss the significant labor time-costs involved with constructing the model and expertise required in 3D modelling and printing. Almost 5 hours of time of a skilled craftsman is not insignificant and this does not include modeling and design.

Overall discussions comments:

- Would discuss limitation of this study further: need for performance comparators to traditional teaching methods/tools. This study used only a simple model type, so results may be specific to this exact 3D printed model, etc…

- Would discuss need for future work in terms of improvements in the model and other advantages 3D printing could over (ie flexibility to create a wider variety of different anatomies, low cost to increase accessibility to learners etc.)

Figure 3, 4, 5: Consider labeling X-axis with actualy question wording if it does not make the figure too crowded.

Figure 6: Consider providing an acutaly comparison ultrasound image of a real human spine.

6. PLOS authors have the option to publish the peer review history of their article (what does this mean?). If published, this will include your full peer review and any attached files.

Reviewer #1: Yes: Azad Mashari

Reviewer #2: No

---

## [Author Response · Author response to Decision Letter 0]

11 Jan 2020

All responses to reviewer and editor comments are contained in the Rebuttal letter.

Submitted as file type "Response to Reviewers"

---

## [Decision Letter · Decision Letter 1]

22 Jan 2020

A do-it-yourself 3D-printed thoracic spine model for anesthesia resident simulation

PONE-D-19-29404R1

Dear Dr. Han,

We are pleased to inform you that your manuscript has been judged scientifically suitable for publication and will be formally accepted for publication once it complies with all outstanding technical requirements.

With kind regards,

Matthew John Meyer, M.D.

Academic Editor

PLOS ONE

Additional Editor Comments (optional):

Thank you for the revision and the focus on differentiating this model from others that are available. I am happy to accept this manuscript for publication. Please address the few issues from Dr. Kuo and the following clarification from me. Thank you and please continue the hard work.

4-87: Please explain why these levels were chosen: "Thoracic vertebrae from T7 to T11 were identified as the most useful for a realistic training model"

Reviewers' comments:

Reviewer's Responses to Questions

**Comments to the Author**

1. If the authors have adequately addressed your comments raised in a previous round of review and you feel that this manuscript is now acceptable for publication, you may indicate that here to bypass the “Comments to the Author” section, enter your conflict of interest statement in the “Confidential to Editor” section, and submit your "Accept" recommendation.

Reviewer #2: All comments have been addressed

2. Is the manuscript technically sound, and do the data support the conclusions?

Reviewer #2: Yes

3. Has the statistical analysis been performed appropriately and rigorously? 

Reviewer #2: N/A

4. Have the authors made all data underlying the findings in their manuscript fully available?

Reviewer #2: Yes

5. Is the manuscript presented in an intelligible fashion and written in standard English?

Reviewer #2: Yes

6. Review Comments to the Author

Reviewer #2: Minor stylistic comments

Abstract: first 6 words of the abstract refer to CVL, cricothyroidtomy and lumbar epidural that have little relevance to the paper. Consider just starting with just "Epidural placements..."

Pg 4 Ln 74: In the purpose statement, also consider introducing the Survey portion of the study by adding something like "...model was developed and attitudes towards the model was surveyed..."

Pg 5 Ln 101: Consider more specific description than "bubble tea straw" which all readers may not be familiar with.

Pg 7 Results, Cost and Fabrication Time: Consider also mentioning cost of the 3D Printer.

Pg 10 Ln 211: Typo, "Expert's" should be "Experts"

7. PLOS authors have the option to publish the peer review history of their article (what does this mean?). If published, this will include your full peer review and any attached files.

Reviewer #2: Yes: Alexander Kuo

---

## [Editor Report · Acceptance letter]

25 Feb 2020

PONE-D-19-29404R1 

A do-it-yourself 3D-printed thoracic spine model for anesthesia resident simulation 

Dear Dr. Han:

I am pleased to inform you that your manuscript has been deemed suitable for publication in PLOS ONE. Congratulations! Your manuscript is now with our production department. 

With kind regards,

on behalf of

Dr. Matthew John Meyer 

Academic Editor

PLOS ONE